# VALUE-DRIVEN HINDSIGHT MODELLING

## ABSTRACT

Value estimation is a critical component of the reinforcement learning (RL) paradigm. The question of how to effectively learn predictors for value from data is one of the major problems studied by the RL community, and different approaches exploit structure in the problem domain in different ways. Model learning can make use of the rich transition structure present in sequences of observations, but this approach is usually not sensitive to the reward function. In contrast, model-free methods directly leverage the quantity of interest from the future but have to compose with a potentially weak scalar signal (an estimate of the return). In this paper we develop an approach for representation learning in RL that sits in between these two extremes: we propose to learn what to model in a way that can directly help value prediction. To this end we determine which features of the *future* trajectory provide useful information to predict the associated return. This provides us with tractable prediction targets that are directly relevant for a task, and can thus accelerate learning of the value function. The idea can be understood as reasoning, in hindsight, about which aspects of the *future* observations could help *past* value prediction. We show how this can help dramatically even in simple policy evaluation settings. We then test our approach at scale in challenging domains, including on 57 Atari 2600 games.

## 1 INTRODUCTION

Consider a baseball player trying to perfect their pitch. The player performs an arm motion and releases the ball towards the batter, but suppose that instead of observing where the ball lands and the reaction of the batter, the player only gets told the result of the play in terms of points or, worse, only gets told the final result of the game. Improving their pitch from this experience appears hard and inefficient, yet this is essentially the paradigm we employ when optimizing policies in model-free reinforcement learning. The scalar feedback that estimates the return from a state (and action), encoding *how well things went*, drives the learning while the accompanying observations that may explain that result (e.g. flight path of the ball or the way the batter anticipated and struck the incoming baseball) are ignored. To intuitively understand how such information could help value prediction, consider a simple discrete Markov chain $X \to Y \to Z$, where $Z$ is the scalar return and $X$ is the observation from which we are trying to predict $Z$. If the space of possible values of $Y$ is smaller than $X$, then it may be more efficient to estimate both $P(Y|X)$ and $P(Z|Y)$ rather than directly estimating $P(Z|X)$.[1] In other words observing and then predicting $Y$ can be advantageous to directly estimating the signal of interest $Z$. Model-based RL approaches would duly exploit the observed $Y$ (by modeling the transition $Y|X$), but $Y$ would, in general scenarios, contain information that is irrelevant to $Z$ and hard to predict. Building a full high-dimensional predictive model to indiscriminately estimate all possible future observations, including potentially chaotic details of the ball trajectory and the spectators' response, is a challenge that may not pay off if the task-relevant predictions (e.g., was the throw accepted, was the batter surprised) are error-ridden. Model-free RL methods directly consider the relation $X$ to $Z$, and focus solely upon predicting and optimising this goal, rather than attempting to learn the full dynamics. These methods have recently dominated the literature, and have attained the best performance in a wide array of complex problems with high-dimensional observations (Mnih et al., 2015; Schulman et al., 2017; Haarnoja et al., 2018; Guez et al., 2019).

---

[1] In the discrete case, this follows from a counting argument from the size of the probability tables involved.

In this paper, we propose to augment model-free methods with a lightweight model of future quantities of interest. The motivation is to model only those parts of the future observations ($Y$) that are needed to obtain better value predictions. The major research challenge is to learn, from observational data, which aspects of the future are important to model (i.e. what $Y$ should be). To this end, we propose to learn a special value function *in hindsight* that receives future observations as an additional input. This learning process reveals features of the future observations that would be most useful for value prediction (e.g. flight path of the ball or the reaction of the batter), if provided by an oracle. These important features are then predicted, in advance, using only information available at test time (at the time of releasing the baseball, we knew the identity of the batter, the type of throw and spin given to the ball). Learning these value-relevant features can help representation learning for an agent and provide an additional useful input to its value and policy. Experimentally, hindsight value functions surpassed the performance of model-free RL methods in a challenging association task (Portal Choice). When hindsight value functions were added to the prior state-of-the-art RL method for Atari games, they significantly increased median performance from 833% to 965%.

## 2 BACKGROUND AND NOTATION

We consider a reinforcement learning setting whereby an agent learns from interaction in a sequential decision-making environment (Sutton & Barto, 2011). An agent's policy $\pi$, mapping states to an action distribution, is executed to obtain a sequence of rewards and observations as follows. At each step $t$, after observing state $s_t$, the policy outputs an action $a_t$, sampled from $\pi(A|s_t)$, and obtains a scalar reward $r_t$ and the next-state $s_{t+1}$ from the environment. The sum of discounted rewards from state $s$ is the return denoted by $G = \sum_{t=0}^{\infty} \gamma^t R_t$, with $\gamma < 1$ denoting the discount factor. Its expectation, as a function of the starting state, is called the value function, $v^\pi(s) = \mathbb{E}_\pi[G|S_0 = s]$. An important related quantity is the action-value, or Q-value, which corresponds to the same expectation with a particular action executed first: $q^\pi(s, a) = \mathbb{E}_\pi[G|S_0 = s, A_0 = a]$. The learning problem consists in adapting the policy $\pi$ in order to achieve a higher value $v^\pi$. This usually entails learning an estimate of $v^\pi$ for the current policy $\pi$, this is the problem we focus on in this paper.

Note that in practice we are interested in partially-observed environments where the state of the world is not directly accessible. For this case, we can think of replacing the observed state $s$ in the case of the fully-observed case by a learned function that depends on past observations.

## 3 VALUE LEARNING

### 3.1 DIRECT LEARNING

A common approach to estimate $v$ (or $q$) is to represent it as a parametric function $v_\theta$ (or $q_\theta$) and directly update its parameters based on sample returns of the policy of interest. Value-based RL algorithms vary in how they construct a value target $Y$ from a single trajectory. They may regress $v_\theta$ towards the Monte-Carlo return ($Y_t = G_t$), or exploit sequentiality in the reward process by relying on a form of temporal-difference learning to reduce variance (e.g. the TD(0) target $Y_t = R_t + \gamma v_\theta(S_{t+1})$). For a given target definition $Y$, the value loss $\mathcal{L}_v$ to derive an update for $\theta$ is: $\mathcal{L}_v(\theta) = \frac{1}{2}\mathbb{E}_s[(v_\theta(s) - Y)^2]$. In constructing a target $Y_t$ based on a trajectory of observations and rewards from time $t$, the observations are either unused (for a Monte Carlo return) or only indirectly exploited (when bootstrapping to obtain a value estimate). In all cases, the trajectory is distilled into a scalar signal that estimates the return of a policy, and other relevant aspects of future observations are discarded. In particular in partially observed domains or domains with high-dimensional observation spaces it can be difficult to discover correlations with this noisy signal.

### 3.2 MODEL-BASED APPROACH

An indirect way of estimating the value is to first learn a model of the dynamics. For example a 1-step observation model $m_\theta$ learns to predict the conditional distribution $s_{t+1}, r_t|s_t, a_t$. Then a value estimate $v(s)$ for state $s$ can be obtained by autoregressively rolling out the model (until the end of the episode or to a fixed depth with a parametric value bootstrap).

The model is trained on potentially much richer data than the return signal since it exploits all information in the trajectory. Indeed, the observed transitions between states can reveal the structure behind a sparse reward signal. A drawback of classic model-based approaches is that they predict a high-dimensional signal, a task which may be costly and harder than directly predicting the scalar value . As a result, the approximation of the dynamics $m_\theta$ may contain errors where it matters most for predicting the value (Talvitie, 2014). Although the observations carry all the data from the environment, most of it is not essential to the task (Gelada et al., 2019). The concern that modeling all observations is expensive also applies when the model is not used for actual rollouts but merely for representation learning. So while classic model-based methods fully use this high-dimensional signal at some cost, model-free methods take the other extreme to focus only on the most relevant low-dimensional signal (the scalar return). Below we propose a method that strikes a balance between these paradigms.

## 3.3 HINDSIGHT VALUE AND MODEL

We introduce a new value function estimate that can only be computed at training time, the *hindsight* value function $v^+$. This value still estimates the expected return from a state $s_t$ but it is further conditioned on $k$ additional observations $\tau_t^+ = s_{t+1}, s_{t+2}, \ldots s_{t+k}$ occurring after time $t$:[2]

$$v^+(s_t, \tau_t^+) \approx \mathbb{E}[G|S_0 = s_t, \ldots, S_k = s_{t+k}]. \tag{1}$$

Furthermore, we require $v^+$ to follow this particular parametric structure:

$$v^+(s_t, \tau_t^+; \theta) = \psi_{\theta_1}(f(s_t), \phi_{\theta_2}(\tau_t^+)), \tag{2}$$

where $\theta = (\theta_1, \theta_2)$, which forces information about the future trajectory through some vector-valued function $\phi \in \mathcal{R}^d$. Intuitively, $v^+$ is estimating the expected return from a past time point using privileged access to future observations. Note that if $k$ is large enough, then $v^+$ simply estimates the empirical return from time $t$ given access to the state trajectory. However, if $k$ is small and $\phi$ is low-dimensional, then $\phi$ becomes a bottleneck representation of the future trajectory $\tau_t^+$. By learning in hindsight, we identify features that are maximally useful to predict the return on the trajectory from time $t$. The hindsight value function is not a useful quantity by itself, since – because of its use of privileged future observations – we cannot readily use it at test time. Furthermore, it cannot be used as a baseline either, as when computing the policy gradient it will yield a biased gradient estimator. . Instead, the idea is to learn a model $\hat{\phi}$ of $\phi$, that can be used at test time. We conjecture that if privileged features $\phi$ are useful for estimating the value, then the model of those features will also be useful for estimating the value function. We propose to learn the approximate expectation model $\hat{\phi}_{\eta_2}(s)$ conditioned on the current state $s$ and parametrized by $\eta_2$, minimizing the following squared loss:

$$\mathcal{L}_{\text{model}}(\eta_2) = \mathbb{E}_{s,\tau^+}[\|\phi_{\theta_2}(s, \tau^+) - \hat{\phi}_{\eta_2}(s)\|_2^2] \tag{3}$$

where the expectation is taken over the distribution of states and partial trajectories $\tau^+$ resulting from that state.

The approximate model $\hat{\phi}$ can then be leveraged to obtain a better model-based value estimate $v^m(s; \eta) = \psi_{\eta_1}(f(s), \hat{\phi}_{\eta_2}(s))$. Although $\hat{\phi}(s)$ cannot contain more information than included already in the state $s$, it can still benefit from having being trained using a richer signal before the value converges. Figure 3 summarizes the relation between the different quantities.

## 3.4 ILLUSTRATIVE EXAMPLE

We consider the following example to illustrate how the approaches to estimating the value function can differ. There are no actions in this example[3] and each episode consists of a single transition from initial state $s$ to terminal state $s'$, with a reward $r(s, s')$ on the way.

Each instance of this example is parametrized by a square matrix $W$ and a vector $b$ sampled from a unit normal distribution, which determine the uncontrolled MDP. Initial states $s$ are of dimension $D$ and sampled from a multivariate unit normal distribution ($s_i \sim N(0, 1)$ for all state dimension $i$).

---

[2]In general $\tau_t^+$ can be defined to include full future transitions, actions and rewards.

[3]This can be understood as a Markov Reward Process or a policy evaluation setting

Given $s = \left(\begin{smallmatrix} s_1 \\ s_2 \end{smallmatrix}\right)$, where $s_1$ and $s_2$ are of dimension $D_1$ and $D_2$ ($D = D_1 + D_2$), the next state $s' = \left(\begin{smallmatrix} s_1' \\ s_2' \end{smallmatrix}\right)$ is determined according to the transition function: $s_1' = \text{MLP}(s) + \epsilon$ and $s_2' = \sigma(W s_2 + b)$ where $\sigma$ is the Heaviside function. $s_1'$ acts as a distractor here, with additive noise $\epsilon \sim N(0, 1)$. The reward obtained is $r(s, s') = \sum_i s_1^{(i)} \sum_i s_2'^{(i)} / \sqrt{D}$. The true value in the start state is also $v(s) = r(s, s')$.

The key aspect of this domain is that $s'$ reveals structure that helps predict the value function in the start state $s$. This is made visually obvious in the trajectories sampled in this domain shown in Figure 1.

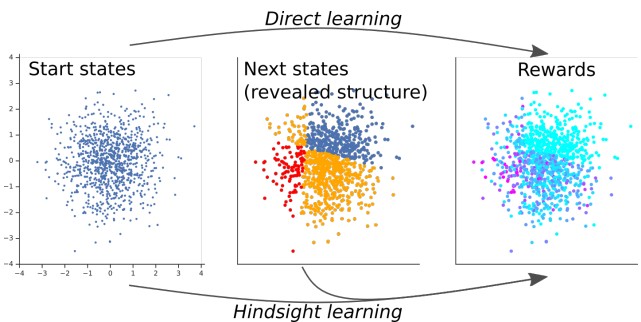

Figure 1: Visualization of episodes in the illustrative example of Section 3.4. Model-free value prediction see the start state on the left and must predict the corresponding color-coded reward on the right. Hindsight value prediction can leverage the observed structure in the intermediate state to obtain a better value prediction. In more detail, this plot shows the second half $s_2$ of initial state $s$ on the left. In the middle, superimposed is the observed reward-relevant quantity $\sum_i s_2'^{(i)}$ that has been color-coded on the $s_2$ vectors. On the right is the color-coded reward for each trajectory. The dimension of states is $D = 4$ in this example.

Let us consider how the different approaches to learning values presented above fare in this problem. For direct learning, the value from $v(s')$ is 0 since $s'$ is terminal, so any n-step return is identical to the Monte-Carlo return, that is, the information present in observation $s'$ is not leveraged. Results from learning $v$ from $s$ given the return is presented in Figure 2 (blue curve). A model-based approach first predicts $s'$ from $s$, then attempts to predict the value given $s$ and the estimated next state. When increasing the input dimension, given a fixed capacity, the model does not focus its attention on the reward-relevant structure in $s'$ and makes error where it matters most. As a result, it can struggle to learn $v$ faster than a model-free estimate (cf. red curve in Figure 2). When learning in hindsight, $v^+$ can directly exploit the revealed structure in the observation of $\tau^+$, and as a result the hindsight value learns faster than the regular causal model-free estimate (cf. dotted yellow curve in Figure 2). This drives the learning of $\phi$ and its model $\hat{\phi}$, which directly gets trained to predict these useful features for the value. As a result, $v^m$ also benefits and learns faster than the regular $v$ estimate on this problem (cf. green curve in Figure 2).

## 3.5 WHEN IS IT ADVANTAGEOUS TO MODEL IN HINDSIGHT?

To understand the circumstances in which hindsight modelling provides a better value estimate, we first consider an analysis that relies on the following assumptions. Suppose that $v_\theta^m$ is sharing the same function $\psi$ as $v^+$ (i.e., $\theta_1 = \eta_1$), and let $\psi$ be linear. If we write $\psi_{\theta_1}(f, \phi) = \left(\begin{smallmatrix} \omega_1 \\ \omega_2 \end{smallmatrix}\right)^\top \left(\begin{smallmatrix} f \\ \phi \end{smallmatrix}\right) + b$, where $\theta_1 = (\omega_1, \omega_2)$, then we have for fixed values of the parameters:

$$\mathbb{E}[(v^m(s; \eta) - v^+(s, \tau^+; \theta))^2] = \mathbb{E}[\|\omega_2^\top (\phi(\tau^+; \theta_2) - \hat{\phi}(s; \eta_2))\|^2] \tag{4}$$

$$\leq \mathbb{E}[\|\omega_2\|^2 \|\phi(\tau^+) - \hat{\phi}(s)\|^2] \tag{5}$$

$$= \|\omega_2\|^2 \mathcal{L}_{\text{model}}(\eta_2), \tag{6}$$

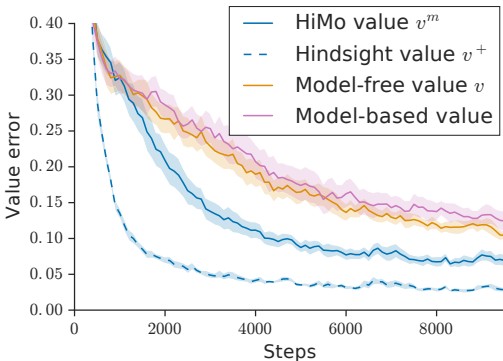

Figure 2: Learning the value of the initial state in the example of Section 3.4. The dimension of the data is $D = 32$ for this experiment, with the dimension of the useful data in the next state $D_2 = 4$. The results are averaged over 4 different instances, each repeated twice. Note that $v^+$ (dotted line) is using privileged information (the next state).

using the Cauchy-Schwarz inequality. Let $\mathcal{L}$ define the value error for a particular value function $v$: $\mathcal{L}(v) = \mathbb{E}[(v(s) - G)^2]$ and $\mathcal{L}(v^+) = \mathbb{E}[(v^+(s, \tau^+) - G)^2]$. Then we have:

$$\mathcal{L}(v^m) = \mathbb{E}[(v^m(s) - v^+(s, \tau^+) + v^+(s, \tau^+) - G)^2] \qquad (7)$$

$$\leq 2(\|\omega_2\|^2 \mathcal{L}_{\text{model}}(\eta_2) + \mathcal{L}(v^+)), \qquad (8)$$

using the fact that $\mathbb{E}[(X + Y)^2] \leq 2(E[X^2] + E[Y^2])$ for random variables $X$ and $Y$. If we assume $\mathcal{L}(v^+) = C\mathcal{L}(v)$ with $C < 0.5$ (i.e., estimating the value in hindsight with more information is an easier learning problem), then the following holds:

$$\mathcal{L}_{\text{model}}(\eta_2) < \frac{(1 - 2C)\mathcal{L}(v)}{2\|\omega_2\|^2} \implies \mathcal{L}(v^m) < \mathcal{L}(v). \qquad (9)$$

In other words, this relates how small the modeling error needs to be to guarantee that the value error for $v^m$ is smaller than the value error for the direct estimate $v$. The modeling error can be large for different reasons. If the environment or the policy is stochastic, then there is some irreducible modeling error for the deterministic model. Even in these cases, a small $C$ can make hindsight modeling advantageous. The modeling error could also be high because predicting $\phi$ is hard. For example, it could be that $\phi$ essentially encodes the empirical return, which means predicting $\phi$ is at least as hard as predicting the value function ($\mathcal{L}_{\text{model}}(\eta_2) \geq \mathcal{L}(v)$). Or it could be that $\phi$ is high-dimensional, this could cause both a hard prediction problem but also would cause the acceptable threshold for $\mathcal{L}_{\text{model}}$ to decrease (since $\|\theta_2\|^2$ will grow). We address some of these concerns with specific architectural choices like $v^+$ having a limited view on future observations and having low dimensional $\phi$ (see next section). Note that the analysis above ignores any advantage that could be obtained from representation learning when training $\hat{\phi}$ (if the state encoding function $f$ shares parameters with $\hat{\phi}$).

## 4 ARCHITECTURE

The architecture for Hindsight Modelling (HiMo) we found to work at scale and tested in the experimental section of the paper is described here. To deal with partial observability, we employ a recurrent neural network, the state-RNN, which replaces the state $s_t$ with a learned internal state $h_t$, a function of the current observation $o_t$ and past observations through $h_{t-1}$: $h_t = f(o_t, h_{t-1}; \eta_3)$, where we have extended the parameter description of $v^m$ as $\eta = (\eta_2, \eta_1, \eta_3)$. The model-based value function $v^m$ and the hindsight value function $v^+$ share the same internal state representation $h$, but the learning of $v^+$ assumes $h$ is fixed (we do not backpropagate through the state-RNN in hindsight). In addition, we force $\hat{\phi}$ to only be learned through $\mathcal{L}_{\text{model}}$, so that $v^m$ uses it as an additional input.

To summarize:

$$v^+(h_t, h_{t+k}; \theta) = \psi_{\theta_1}(\overline{h_t}, \phi_{\theta_2}(\overline{h_{t+k}})), \tag{10}$$

$$v^m(h_t; \eta) = \psi_{\eta_1}(h_t, \overline{\hat{\phi}_{\eta_2}(h_t)}), \tag{11}$$

with the bar notation denoting quantities treated as non-differentiable (i.e. where the gradient is stopped). The different losses in the HiMo architecture are combined in the following way:

$$\mathcal{L}(\theta, \eta) = \mathcal{L}_v(\eta) + \alpha \mathcal{L}_{v^+}(\theta) + \beta \mathcal{L}_{\text{model}}(\eta). \tag{12}$$

A diagram of the architecture is presented in Figure 3, and further implementation details can be found in the appendix.

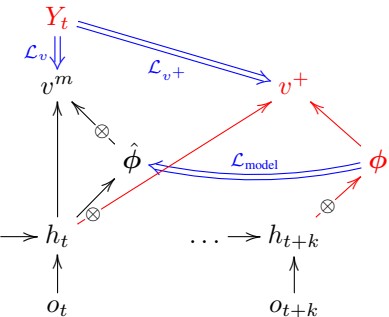

Figure 3: Network architecture for HiMo. Double blue arrows denote losses on different outputs of the network. Red denote quantities which are only computed in hindsight at train time (using parameters $\theta$). The $\otimes$ symbol on an arrow means its input is assumed to be non-differentiable (also sometimes called a stop gradient).

This architecture can be straightforwardly generalized to cases where we also output a policy $\pi_\eta$ for an actor-critic setup, providing $h$ and $\hat{\phi}$ as inputs to a policy network.[4] For a Q-value based algorithm like Q-learning, we predict a vector of values $q^m$ and $q^+$ instead of $v^m$ and $v^+$. Computing $v^+$ and training $\hat{\phi}$ can be done in an online fashion by simply delaying the updates by $k$ steps (just like the computation of an $n$-step return).

## 5 EXPERIMENTS

The illustrative example in Section 3.4 demonstrated the positive effect of hindsight modeling in a simple policy evaluation setting. In this section, we now explore these benefits in the context of policy optimization in challenging domains, a custom navigation task called Portal Choice, and Atari 2600. To demonstrate the generality and scalability of our approach we test hindsight value functions in the context of two high-performance RL algorithms, IMPALA (Espeholt et al., 2018) and R2D2 (Kapturowski et al., 2019).

### 5.1 PORTAL CHOICE TASK

The Portal Choice (Fig. 4) is a two-phase navigation task where, in phase one an agent is presented with a contextual choice between two portals, whose positions vary between episodes. The position of the portal determines its destination in phase two, one of two different goal rooms (green and red rooms). Critically, the reward when terminating the episode in the goal room depends on both the color of the goal room in phase two and a visually indicated combinatorial context shown in the first phase. If the context matches the goal room color, then a reward of $2$ is given, otherwise the reward is $0$ when terminating the episode (see appendix for the detailed description).

An easy suboptimal solution is to select the portal at random and finish the episode in the resulting goal room by reaching the goal pixel, which will result in a positive reward of $1$ on average. A more

---

[4]In this case, the total loss also contains an actor loss to update $\pi_\eta$ and a negative entropy loss.

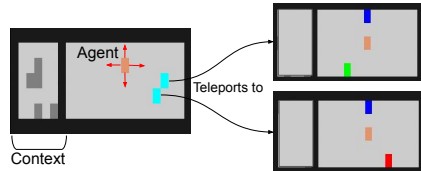

Figure 4: Portal Choice task. Left: an observation in the starting room of the Portal Choice task. Two portals (cyan squares) are available to the agent (orange), each of them leading to a different room deterministically based on their position. Right: The two possible goal rooms are identified by a green and red pixel. The reward upon reaching the goal (blue square) is a function of the room and the initial context.

difficult strategy is to be selective about which portal to take depending on the context, in order to get the reward of 2 at each every episode. A model-free agent has to learn the joint mapping from contexts and portal positions to rewards. Even if the task is not visually complex, the context is combinatorial in nature (the agent needs to count randomly placed pixels) and the joint configuration space of context and portal is fairly large (around 250M). Since the mapping from portal position to rooms does not depend on context, learning the portal-room mapping independently is more efficient in this scenario.

For this domain, we implemented the HiMo architecture within a distributed actor-critic agent, named IMPALA proposed by Espeholt et al. (2018). In this case, the target $Y_t$ to train $v^m$ (used as a critic in this context) and $v^+$ is the V-trace target (Espeholt et al., 2018) to account for off-policy corrections between the behavior policy and the learner policy. The actor shares the same network as the critic and receives $h$ and $\hat{\phi}$ as inputs.

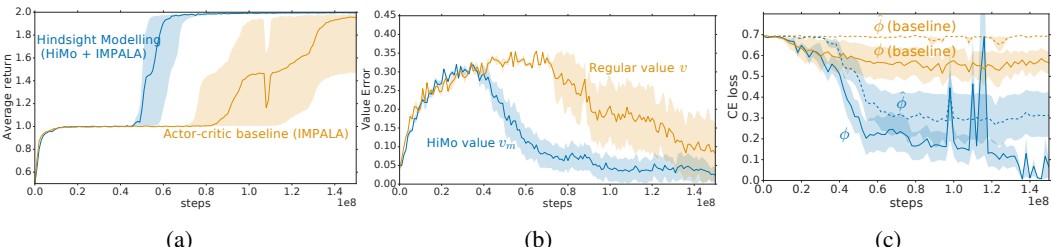

Figure 5: Results in the Portal Choice task. (a) shows the median performance as a function of environment steps out of 4 seeds. (b) shows the value error averaged across states on the same x-axis scale for different value function estimate. (c) is an analysis that shows the cross-entropy loss of a classifier that takes as input $\phi$ (solid line) or $\hat{\phi}$ (dotted line) and predicts the identity of the goal room (red or green) as a binary classification task. The HiMo curves (blue) show that information about the room identity becomes present first in $\phi$ and then gets captured in its model $\hat{\phi}$. For the baseline (where we set $\alpha = \beta = 0$), $\hat{\phi}$ is not trained based on $\phi$ and only achieves to classify the room identity at chance level.

We found that HiMo+IMPALA learned reliably faster to reach the optimal behavior, compared to the vanilla IMPALA baseline that shared the same network capacity (see Figure 5a). The hindsight value $v^+$ rapidly learns to predict whether the portal-context association is rewarding based on seeing the goal room color in the future. Then $\phi$ learns to predict the new information from the future that helps that prediction: the identity of the room (see analysis Fig 5c). The prediction of $\phi$ becomes effectively a model of the mapping from portal to room identity (since the context does not correlate with the room identity). Having access to such mapping through $\hat{\phi}$ helped the value prediction (Fig 5b), which led to better action selection. Note that if the two rooms were visually indistinguishable, for example with no red/green rooms separation, HiMo would not be able to offer any advantage over its model-free counterpart.

## 5.2 ATARI

We tested our approach in Atari 2600 videogames using the Arcade Learning Environment (Bellemare et al., 2013). We added HiMo on top of Recurrent Replay Distributed DQN (R2D2), a DQN-

based distributed architecture introduced by Kapturowski et al. (2019) which achieved state-of-the-art scores in Atari games.

In this value-based setting, HiMo trains $q^m(\cdot, \cdot; \eta)$ and $q^+(\cdot, \cdot; \theta)$ based on $n$-step return targets:

$$Y_t = g\left(\sum_{m=0}^{n-1} \gamma^m R_{t+m} + \gamma^n g^{-1}\left(q^m(S_{t+n}, A^*; \eta^-)\right)\right), \tag{13}$$

where $g$ is an invertible function, $\eta^-$ are the periodically updated target network parameters (as in DQN by Mnih et al. (2015)), and $A^* = \arg\max_a q^m(S_{t+n}, a; \eta)$ (the Double DQN update proposed by Van Hasselt et al. (2016)). The details of the architecture and hyperparameters are described in the appendix.

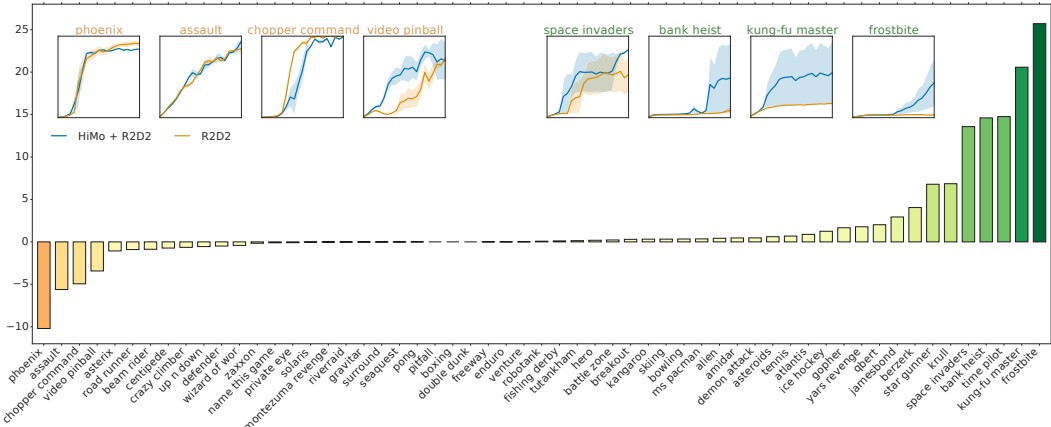

Figure 6: Difference in human normalized score per game in Atari, HiMo versus the improved R2D2 after 200k learning steps, alongside learning curves for a selection of HiMo worst and top performing games. Note that the high variance of the curves in Atari between seeds can usually be explained by the variable timestep at which different seeds jump from one performance plateau to the next.

We ran our approach on 57 Atari games for 200k gradient steps (around 1 day of training), with 3 seeds for each game. The evaluation averages the score between 200 episodes across seeds, each lasting a maximum of 30 minutes each and starting a random number (up to 30) of no-op actions. In order to compare scores between different games and aggregated results, we computed normalized scores for each game based on random and human performance so that 0% corresponds to random performance and 100% corresponds to human. We observed an increase of 132.5% in the me-

Table 1: Median and mean human normalized scores across 57 Atari2600 games for HiMo versus the R2D2 baseline after a day of training.

|        | R2D2    | R2D2 + HiMo |
|--------|---------|-------------|
| Median | 832.5%  | **965%**    |
| Mean   | 2818.5% | **2980%**   |

dian human normalized score compared to the R2D2 baseline with the same network capacity, aggregate results are reported in Table 1. Figure 6 details the difference in normalized score between HiMo and our R2D2 baseline for all games individually. We note that the original R2D2 results reported by Kapturowski et al. (2019), which used a similar hardware configuration but a different network architecture, were around 750% median human normalized score after a day of training.

In our experimental evaluation we observed that HiMo typically either offers improved data efficiency or has no overwhelming adverse effects in training performance. In Figure 6 we show training curves for a selection of representative Atari environments where at evaluation time HiMo both under-performed (left) and out-performed R2D2 (right); these seem to indicate that in the worst case scenario HiMo's training performance reduces to R2D2's.[5]

Bowling is one of the Atari games where rewards are delayed with relevant information being communicated through intermediate observations (the ball hitting the pins), just like the baseball example

---

[5]We will include a performance analysis over longer training regimes in a future version of the paper.

we have used in the introduction. We found HiMo to perform better than the R2D2 baseline in this particular game. We also ran HiMo with the actor-critic setup (IMPALA) described in the previous section, finding similar performance gain with respect to the model-free baseline. Learning curves for these experiments are presented in Figure 7.

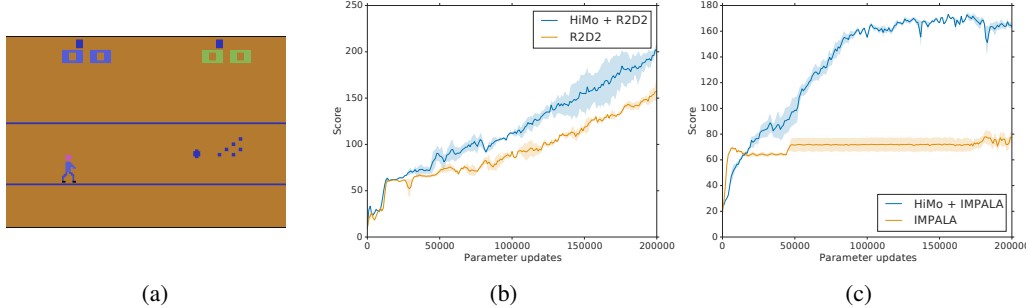

(a)                                     (b)                                     (c)

Figure 7: (a) The bowling game in Atari, where a delayed reward can be predicted by the intermediate event of the ball hitting the pins. (b-c) Learning curves for HiMo in the bowling game using two different RL methods: a value-based method (R2D2) in (b) and a policy-gradient method (IMPALA) in (c).

## 6   RELATED WORK

Recent work have used auxiliary predictions successfully in RL as a mean to obtain a richer signal for representation learning (Jaderberg et al., 2016; Sutton et al., 2011). However these additional prediction tasks are hard-coded and so they cannot adapt to the task demand when needed. We see them as a complementary approach to more efficient learning in RL.

Buesing et al. (2018) have considered using observations in an episode trajectory in hindsight to infer variables in a structural causal model of the dynamics, allowing to reason more efficiently in a model-based way about counterfactual actions. However this approach requires learning an accurate generative model of the environment.

In supervised learning, the learning using privileged information (LUPI) framework introduced by (Vapnik & Izmailov, 2015) considers ways of leveraging privileged information at train time. Although the techniques developed in that work do not apply directly in the RL setting, some of our approach can be understood in that setting as considering the future trajectory as the privileged information for a value prediction problem.

Privileged information coming from full state observation has been leveraged in RL to learn better critic in asymmetric actor-critic architectures (Pinto et al., 2017; Zhu et al., 2018). However this does not use future information and only applies to settings where special side-information (full state) is available at train time.

## 7   CONCLUSION

High-dimensional observations in the intermediate future often contain task-relevant features that can facilitate the prediction of an RL agent's final return. We introduced a reinforcement learning algorithm, HiMo, that leverages this insight by the following two-stage approach. First, by reasoning in hindsight, the algorithm learns to extract relevant features of future observations that would be been most helpful for estimating the final value. Then, a forward model is learned to predict these features, which in turn is used as input to an improved value function, yielding better policy evaluation and training at test time. We demonstrated that this approach can help tame complexity in environments with rich dynamics at scale, yielding increased data efficiency and improving the performance of state-of-the-art model-free architectures.

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

# A APPENDIX

## A.1 GENERAL ARCHITECTURE DETAILS

To compute $v^+$ and train $\hat{\phi}$ in an online fashion, we process fixed-length unrolls of the state-RNN and compute the hindsight value and corresponding updates at time $t$ if $t + k$ is also within that same unroll. Also, we update $v^+$ at a slower rate (i.e., $\alpha < \beta$) to give enough time for the model $\hat{\phi}$ to adapt to the changing hindsight features $\phi$. In our experiments we found that even a low-dimensional $\phi$ (in the order of $d = 3$) and a relatively short hindsight horizon $k$ (in the order of 5) are sufficient to yield significant performance boosts, whilst keeping the extra model computational costs modest.

## A.2 PORTAL CHOICE

**Environment** The observation is a $7 \times 23$ RGB frame (see Figure 4). There are 3 possible spawning points for the agent in the center and 42 possible portal positions (half of which lead to the green room, the other half leading to the red room). At the start of an episode, two portals, each leading to a different room, are chosen are random. They are both displayed as cyan pixels. Included in the observation in the first phase is the context, a random permutation in a $5 \times 5$ grid of $N$ pixels, where is uniformly sampled at the start of each episode: $N \sim \mathcal{U}\{1, 10\}$. A fixed map $f : \{1, \ldots, 10\} \to \{0, 1\}$ determines which contexts are rewarding with the green room, the rest being rewarding with the red room. The reward when reaching the goal is determined according to:

$$R = 2(f(N)G + (1 - f(N))(1 - G)), \tag{14}$$

where $G \in \{0, 1\}$ is whether the reached room is green.

**Network architecture** The policy and value network takes in the observation and passes it to a ConvNet encoder (with filter channels [32, 32, 32], kernel shapes [4, 3, 3] applied with strides [2, 1, 1]) before being passed to a ConvLSTM network with 32 channels and 3x3 filters. The output of the ConvLSTM is the internal state $h$. The $\hat{\phi}$ network is a ConvNet with [32, 32, 32, 1] filter channels with kernels of size 3 except for a final 1x1 filter, whose output is flatten and passed to an MLP with 256 hidden units with ReLu activation, before a linear layer with dimension $d = 3$. The $\phi$ network is a similarly configured network with one less convolution layer and 128 hidden units in the MLP. The $\psi_\eta$ network is an MLP with 256 hidden units followed by a linear layer that takes $h$ and $\hat{\phi}$ as input and outputs the policy $\pi^m$ and the value $v^m$. $v^+$ is obtained similarly with a similar MLP that has a single scalar output. We used a future observation window of $k = 5$ steps in this domain and loss weights $\alpha = 0.25$, $\beta = 0.5$. Unroll length was 20, and $\gamma = 0.99$. Optimization was done with the Adam optimizer (learning rate of $5e - 4$), with batch size 32. The model-free baseline is obtained by using the same code and network architecture, and setting the modeling loss and hindsight value loss to 0 ($\alpha = \beta = 0$).

## A.3 ATARI

Hyper-parameters and infrastructure are the same as reported in Kapturowski et al. (2019), with deviations as listed in table 2. For our value target, we also average different $n$-step returns with exponential averaging as in $Q(\lambda)$ (with the return being truncated at the end of unrolls). The $Q$ network is composed of a convolution network (cf. Vision ConvNet in table) which is followed by an LSTM with 512 hidden units. What we refer to in the main text as the internal state $h$ is the output of the LSTM. The $\phi$ and $\hat{\phi}$ networks are MLPs with a single hidden layer of 256 units and ReLu activation function, followed by a linear which outputs a vector of dimension $d$. The $\psi_{\theta_1}$ function concatenates $h$ and $\phi$ as inputs to an MLP with 256 hidden units with ReLu activation function, followed by a linear which outputs $q^+$ (a vector of dimension 18, the size of the Atari action set). $q^m$ is obtained by passing $h$ and $\hat{\phi}$ to a dueling network as described by Kapturowski et al. (2019).

Other HiMo parameters are described in table 3. The R2D2 baseline with the same capacity is obtained by running the same architecture with $\alpha = \beta = 0$.

Table 2: Hyper-parameter values used for our R2D2 implementation.

| | |
|---|---|
| Number of actors | 320 |
| Sequence length | 80 (+ prefix of l = 20 in burn-in experiments) |
| Learning rate | $2e^{-4}$ |
| Adam optimizer $\beta_1$ | 0.9 |
| Adam optimizer $\beta_2$ | 0.999 |
| $\lambda$ | 0.7 |
| Target update interval | 400 |
| Value function rescaling | $g(x) = \text{sign}(x)\left(\sqrt{\|x\| + 1} - 1\right) + \epsilon x,\ \epsilon = 10^{-3}$ |
| Frame pre-processing | None (full res. including no frame stacking) |
| Vision ConvNet filters sizes | [7, 5, 5, 3] |
| Vision ConvNet filters strides | [4, 2, 2, 1] |
| Vision ConvNet filters channels | [32, 64, 128, 128] |

Table 3: Hindsight modeling parameters for Atari

| | |
|---|---|
| $\alpha$ | 0.01 |
| $\beta$ | 1.0 |
| $k$ | 5 |
| $d$ | 3 |

