# OpenReview forum: "Value-Driven Hindsight Modelling"
_ICLR.cc/2020/Conference — Reject_

### Official Review · AnonReviewer2 · 2019-10-21
**Official Blind Review #2**

**Rating:** 6

**Review:**

This paper presents a new model-based reinforcement learning method, termed hindsight modelling. The method works by training a value function which, in addition to depending on information available at the present time is conditioned on some learned embedding of a partial future trajectory. A model is then trained to predict the learned embedding based on information available at the current time-step. This predicted value is fed in place of the actual embedding to the same value model, to generate a value prediction for the current time-step. So instead of just learning a value function based on future returns, the method uses a two-step process of learning an embedding of value relevant information from the future and then learns to predict that embedding.

The paper gives some motivating examples of why and when such an approach could yield an advantage over standard value learning methods like Monte-Carlo or temporal difference learning. The basic idea is that when the returns obey some causal structure like X->Y->Z it may be easier to learn P(Y|X) and P(Z|Y) than to learn P(Z|X) directly. In particular, the authors point out that in the discrete case when Y takes relatively few values the size of the respective probability tables can be smaller in the former case than the latter. This motivates the approach of discovering a set of future variables to predict which are themselves predictive of return, rather than predicting the expected future return directly.

On a high level, I like the idea of specifically learning to model relevant aspects of the environment. However, I lean toward rejecting this work in its current form because I feel the motivation for when this particular method would be useful is unclear.

In particular, I don't really understand how this method could, in general, be expected to improve on regular bootstrapping. Why learn a prediction of the return at time t based on future information when we could just use the value function at a later time to improve the prediction at time t? It seems to me that the future value function itself concisely summarizes the information in the future state that is relevant for predicting the past return, while better exploiting the structure of the problem. Of course in cases like partial observability, it could be that the future value function lacks information from the past that is important for accurately predicting the return (for example in the portal example of this paper). However, if partial observability is really the case of interest, the method presented in this paper seems like a rather roundabout solution method. For example, instead of conditioning v+ on a future hidden state h_{t+k} (as the authors do in the experiments) perhaps one could simply condition the value function on a past hidden state h_{t-k} and obtain similar benefit from bootstrapping?

Aside from partial observability, for which I feel there are better approaches, the only situation I can understand the method having an advantage is when later states contain information which helps to predict earlier rewards. This is essentially the situation presented in the illustrative example. However, currently I feel such situations are rather contrived and unintuitive so I would need more supporting evidence to accept these situations as a good motivation.

On a deeper level, I don't see how the probability table motivation given in the introduction applies when what is being learned is an expectation (i.e. a value function) and not a distribution.

The approach also suffers from well-known issues with using the output of an expectation model of a variable as the input to a nonlinear function approximator in place of the variable itself. Namely, there is no guarantee that the expectation value of a variable is a possible value for the variable so giving it as input to a predictor trained on the variable itself could easily yield nonsense output in the stochastic case. As far as I can tell the method does nothing to mitigate this (please correct me if I'm wrong), so there is no reason to assume the method is generally applicable in settings with nontrivial stochasticity.

Despite these concerns, I feel the experiments are for the most part quite well thought out and executed. The paper is also quite well written, motivation issues aside, so I would not be upset if it was accepted with the hope that it leads to future work addressing the above-mentioned concerns.

If possible I think this paper would benefit significantly from a detailed explanation of how and when the proposed approach should be expected to improve on bootstrapping, including bootstrapping off a value function which uses an analogous architecture to v+.

Questions for authors:
Given the hyper-parameters of R2D2 deviate somewhat from those used in the original paper, and nothing is said about how they were chosen, how confident can we be that the observed advantage of hindsight modelling is not simply due to hyper-parameters being selected which are more favourable for the proposed method?

Given that you are not learning distributions but expectations in the form of value functions, how pertinent is the motivation of learning P(Y|X) and P(Z|Y) instead of P(Z|X) directly described in the introduction?

How much of the benefit observed in the portal example an ATARI could also be gained from simply providing the value function approximation with h_{t-k} as input to help span larger time-gaps?

Update:

While I still feel the exposition could be improved to make the underlying idea clearer, I feel the authors did a good job of addressing my major concerns in their reply, hence I have raised my score to a weak accept.

I have to admit I missed the point that v^+ and v^m were using entirely different parameter sets. In light of this, I agree that the expectation model issue I mentioned is not a major concern.

I also appreciate the clarification of the hyperparameters, if they were really tuned to improve the baseline then this detail should be added to the paper and would negate my concern there.

Finally, I thank the authors for providing the value-function oriented example. I found this example to be more illustrative than the one in the introduction of the paper, and I now feel that I have a better grasp of the motivation. I still have doubts about the general benefit of the approach over bootstrapping but since it is not entirely clear to me one way or the other I feel the idea at least warrants further exploration, and it would be reasonable to accept the paper to make the community aware of it.


**Experience Assessment:**

I have published one or two papers in this area.

**Review Assessment: Checking Correctness Of Derivations And Theory:**

I assessed the sensibility of the derivations and theory.

**Review Assessment: Checking Correctness Of Experiments:**

I assessed the sensibility of the experiments.

**Review Assessment: Thoroughness In Paper Reading:**

I read the paper thoroughly.

---

> ### Author Response · Authors · 2019-11-14
> **Response to Review #2**
>
> Thank you for taking the time to review our paper.
>
> Let us clarify the two major concerns first:
>
> RE: About bootstrapping:
>
> We tried to convey the difference in Section 3.1 but we will add more clarification there since this is an important and subtle point.
>
> Bootstrapping only helps to provide potentially better value targets from a trajectory (e.g., to reduce variance or deal with off-policyness), but it does not give a richer training signal. It does not communicate more information about the future than a return statistic. Here is a simple example to make this concrete, consider a policy evaluation problem in a deterministic scenario where we want to estimate v(s_0) (value at time 0), and v_theta(s_t) = v(s_t) for t > 0 (i.e., all subsequent values are perfect). Then Monte-Carlo returns g from start states and n-step returns are equal: G = R_0 + \gamma v_theta(s_1) = R_0 + \gamma R_1 + \gamma^2 v_(s_2) = …
> So whether you choose to bootstrap or not has no consequence in this scenario (the value target will be the same), yet there might still be some useful information in the trajectory, say in s_2, which if we could predict from s_0 would accelerate the learning of v_theta(s_0). HiMo is capable of leveraging this information because these features in s_2 would be useful for the hindsight value function.
>
> This effect is not restricted to deterministic problems. In fact, the illustrative example (section 3.4) also conveys that point and it is stochastic. There, the MDP is just composed of a single transition, so bootstrapping is not applicable. Yet hindsight modeling can still extract valuable information from that single transition and learn V faster, as we show in Figure 2.
>
>
> RE: About expectation (and prob. Table example):
>
> The probability table was meant to be understood at some intuitive level as a motivational statement since it is easy to understand the counting argument, and we thought was appropriate therefore in the introduction.  A very similar argument for a deterministic mapping can be made which then applies more directly to a value function. Consider the following chain:
>
> (X, X’) -> (X,Y’) -> Z
>
> Z is to be interpreted as the expected return. Here the start state (X,X’) is sampled randomly but the rest of the chain has deterministic transition, and Y’ is independent of X given X’. Let N = number of possible values of X, M = number of possible values of X’, and suppose the number of possible values of Y’ is 2. In a tabular setting, learning the start state value function “model-free” (mapping (X,X’) directly to Z) requires observing returns Z for all NM entries. In contrast, if we estimate the mappings X’->Y’ and (X,Y’)->Z separately this requires M + 2N entries, which is better than NM (for N,M > 4).
>
> Regarding the more general point about using an expectation model. It’s true that in the stochastic case there will be some irreducible model error, but the expected \phi is still a valid and potentially useful statistic of the future. Moreover, we want to emphasize that we use a different parametrization for the hindsight (v^+) and model-augmented (v^m) value networks in the experiments (that is \theta_1 != \eta_1). So even if v^m is conditioned on the expected estimate of the hindsight feature \phi, the network can learn to interpret that statistic (and therefore we don’t run into the issue you mention). That being said, the more general idea we are putting forward in this paper is not incompatible with having a stochastic model.
>
> RE: the hyperparameters
>
> R2D2 with the parameters from the original paper perform less well, so these parameters are advantageous to both methods (Parameters had been selected to improve the baseline R2D2 performance). The control experiment runs exactly the same architecture with some losses set to 0 to control for capacity and speed, so we believe this is a fair controlled experiment.
>
> RE: the proposal of giving h_{t-k} as input
>
> The network in our experiments is recurrent, so providing past information as additional input would only help if there was some memory requirement that the network is not able to satisfy.
> In the portal task, the important decision doesn’t require any memory (everything is observed in the portal room to select the portal), but there is a memory demand when in the reward room. We ran an extra experiment where we gave h_{t-k} as an additional input to the policy and value for the baseline and it did not perform better than what is reported in Figure 5a for the actor-critic baseline.

---

### Official Review · AnonReviewer1 · 2019-10-23
**Official Blind Review #1**

**Rating:** 6

**Review:**

Summary:

The paper proposes a way to learn better representation for RL by employing a hindsight model-based approach. The reasoning is that during training, we can observe the future trajectory and use features from it to better predict past values/returns. However to make this practical, the proposed approach fits an approximator to predict these features of the future trajectory from the current state and then subsequently, use them to predict the value. The authors claim that this extra information can be used to learn a better representation in some problems and lead to faster learning of good policies (or optimal value functions)

Decision:
Weak Accept
My decision is influenced by the following two key reasons

(1) I like the idea of hindsight modeling a lot. It is true that a trajectory gives much more information than just a weak scalar signal indicating return from each state in the trajectory. Identifying a way to make use of all the extra information in the trajectory to aid in value prediction is useful. The proposed approach is a step towards that, and I think the community should be made aware of that for sake of future research in this direction.

(2) Having said that, I am not super satisfied with the way the authors have presented their approach. The explanation is jumbled and confusing, at times. The paper needs careful rewriting to communicate ideas better and notation needs to be standardized earlier. Some of the sections are either redundant or lack insights. Even if they do have insights, they are not highlighted leaving the reader to search for them. The experimental setup is not clear and the authors could have spent more space in the paper dedicated to how the hindsight modeling approach can be implemented within an existing RL method.

Comments:

(1) The line in abstract "but this approach is usually not sensitive to reward function" doesn't make sense. Isn't reward function part of the model? So you are learning the reward function, so how is it not sensitive to it? I think I understand what the authors are saying but it took me until the end of Sec 3.2 to get that.

(2) How does this work relate to Value-aware model learning works from Farahmand (AISTATS 2017, NeurIPS 2018). The premise seems to be similar: learn a model taking into account the underlying decision-making problem to be solved and the structure of the value function. The paper needs a discussion of these set of works

(3) In Section 3.3, \phi_{\theta_2} has conflicting function parameters in eq (2) and (3).

(4) Section 3.4 is very confusing. I understood the setup of the problem and it seemed like it was very illustrative of an example where proposed approach will excel. However, Fig 1 and its caption are unclear and I found it hard to understand what the figure is conveying. The paragraph underneath the figure had no explanation for the Fig 1, and instead directly jumped to the results in Fig 2. The paper could use a better explanation of Fig 1. and explain why the proposed approach can learn the structure of s' and better predict value at s

(5) Section 3.5 partially answers the question "when is it advantageous to model in hindsight?" In cases, where L_model is low, of course its advantageous to model in hindsight! But the real question that needs to be answered is buried in the last paragraph. What if learning a good \phi is as hard as predicting the return? In this case, do we still gain any advantage? I am not sure how having a limited view of future observations and low dimensional \phi helps. If the feature that decides future return lies beyond the limited view of future observations, does it still not give any advantage? Questions like these might be useful in aiding the reader to understand why hindsight modeling is better

(6) Section 4 needs more text to explain what components of the architecture are learnt using what losses, and provide intuitions for why that is the case. It seems like that is very crucial to ensure that \phi doesn't learn something trivial and non-useful. I am surprised section 4 is so small, and Fig 3 is not useful. Maybe, you can combine section 3.4, 3.5 and condense them, and using the obtained space in expanding sec 4.

(7) The experiments section immediately dives into the problem setup and results. It will be useful to have a subsection explaining how the proposed hindsight model is implemented within an RL algorithm. Currently, it is hard for the reader to connect what he/she has read until Section 4 with what's presented in Section 5.

(8) The results are convincing. However, my biggest concern is the experiments were not designed carefully to analyze how much the hindsight modeling contributed in the increase of performance? Are the number of parameters in the value function approximator the same between the hindsight RL algorithm and the baseline? Can we have a simplistic example that is amenable to isolate the influence of hindsight modeling from other factors? Fig. 2 does a reasonable job at it but I think the hindsight modeling approach can achieve improvement in more diverse problems. In a way, the proposed feature is doing state space augmentation so that value can be easily predicted from the features of the augmented state. So, identifying the characteristics of the problems where this can be done is very useful to the RL practictioner.

**Experience Assessment:**

I have published one or two papers in this area.

**Review Assessment: Checking Correctness Of Derivations And Theory:**

I carefully checked the derivations and theory.

**Review Assessment: Checking Correctness Of Experiments:**

I carefully checked the experiments.

**Review Assessment: Thoroughness In Paper Reading:**

I read the paper at least twice and used my best judgement in assessing the paper.

---

> ### Author Response · Authors · 2019-11-14
> **Response to Review #1**
>
> Thank you for taking the time to review our paper and the detailed and specific feedback.
>
> 1-  Even when the model also predicts the reward, the observation model is usually asked to indifferently predict all of the high-dimensional observation, even if some aspects are not relevant to the task (i.e. to the rewards).  The transition model does not focus on what is most important for the task, so it may not be data-efficient to learn (and it may be limited by capacity).
>
> Nonetheless, we’ll attempt to clarify that specific sentence in the abstract.
>
> 2-  Thanks for pointing out this work, it is indeed relevant but the method is quite different. In the value-aware model learning work (iterative Value-Aware Model Learning), the model loss minimizes some form of value consistency: the difference between the value at some next state and the expected value from starting in the previous state, applying the model and computing the value. While this makes the model sensitive to the value, it only exploits the future real state through V as a learning signal (just like in bootstrapping). In contrast, our model is both sensitive to the value and can exploit a richer signal from the future observations. We’ll discuss that work in the related work section.
>
> 3-  This is a typo, the state shouldn’t be part of \phi_{\theta_2}’s arguments in Eq 3.
>
> 4-  Thanks for the feedback. We will update the legend of Fig 1 and the example description to make it clearer.
>
> 5-7 We are sorry you found the current exposition of the work in these sections to not be ideal. We will attempt to improve the balance between certain sections and especially expand the discussion of the architecture.
>
> 8- In all our experiments, the number of parameters and the computational cost  of evaluating the network is the same in HiMo and the baseline because we use the exact same network architecture and only set some of the extra losses to 0 to obtain the baseline. We proposed two domains (the illustrative task and the portal example) where we have isolated some problem features where hindsight modelling is particularly relevant. We thought it was also important to test it in domains that were not specially conceived to test the idea and we chose the 57 Atari games for that. Of course, more extensive empirical investigations is always desirable but we believe this is sufficient to establish that this novel idea can be successful in practice.

---

> > ### Comment · AnonReviewer1 · 2019-11-15
> > **Response to Authors - Final Decision**
> >
> > I agree that it is an interesting idea and shows promise. However, given the current exposition and investigation done in the paper about the approach, I feel that a 'weak accept' is the right decision for this manuscript. I hope this doesn't deter the authors from working on this in the future, and I hope to see a more polished version of this manuscript out soon :)
> >
> > Thanks!

---

### Official Review · AnonReviewer3 · 2019-10-27
**Official Blind Review #3**

**Rating:** 6

**Review:**

Value-Driven Hindsight Modelling proposes a method to improve value function learning. The paper introduces the hindsight value function which estimates the expected return at a state conditioned on the future trajectory of the agent. How use this hindsight value function is not obvious, since an agent does not have access to the future states needed in order to take actions (for Q-Learning) and the hindsight value function is a biased gradient estimator for training policy gradient methods.

The authors train the standard value function (which does not have access to future information) to predict the features which the highsight value function learns to summarize the value relevant parts of the future trajectory. These predicted features can then be used in place of the actual hindsight value function, circumventing the issues discussed above. The authors argue that this auxiliary objective provides a richer training signal to the normal value function, helping it to better learn what information in a given state is relevant to predicting future rewards.

The paper is well structured and written, flowing from high level motivation and review into the core of the method, followed by analysis of the approach, and then proceeds through three experiments. The first two are toy / crafted experiments which build intuition and probe the behavior of the method and finally a large scale test on the Atari 57 benchmark demonstrating improvements when augmenting a state-of-the-art method with HiMo.

This reviewer recommends acceptance (I would give a 7 given more granularity) based on the contribution of a new auxiliary objective for value functions and the strength of the experimental suite. The Portal Choice environment is well crafted and instrumented with the graphs of figure 5b and 5c to show the behavior of the approach and the clean demonstration of an improvement over a previously SOTA method for Atari 57 is encouraging (the same architecture and the ablation simply sets the auxiliary objective’s weight to 0). However, the reviewer has some caution and concerns as follows:

1) The lack of a large scale experiment demonstrating improvement with an actor-critic method. While the Portal Choice experiments are informative and use Impala, it is a bit toy, and it would increase the reviewer’s confidence in the generality and robustness of the approach if improvements were also demonstrated for an actor-critic method on a large environment suite. Atair 57 could work but ideally a different setting such as DMLab 30 or continuous control from pixels. Demonstrating improvements in one of these additional settings would raise the reviewer to a strong acceptance.

2) The potential sensitivity of the approach to the two important hyperparameters that the authors mention, the dimensionality of the hindsight feature space (to reduce approximation error) and the # of future states it conditions on (to avoid just observing the full return directly). The very low dimensionality of the hindsight feature space (d=3 for Atari) seems a bit at odds with the explanation that the hindsight features provide a strong training signal for learning to better extract value relevant information from the state. Experiments that studied sensitivity to these would provide better perspective on the robustness of HiMo.

Questions and suggestions for improving the paper:

For Figure 6 the dynamic range gets squashed by a few games with relatively large performance improvements or regressions. Changing to a log-scale on the y-axis could be more informative? For instance, I find it pretty difficult to eyeball the ~1 human normalized score median improvement according to Table 1 from the chart.

Figure 3 could also be improved. It requires significant context from definitions in the paper in order to understand. It could be reworked into a stand alone expository overview of HiMo that helps readers quickly grok the idea of the paper such that abstract + figure is enough.

Could the authors consider showing / adding full learning curves (median human normalized score?) for HiMO vs the baseline on Atari 57? This would help readers get a qualitative feel for the learning dynamics of the algorithm instead of only having a final scalar measure at the end of training.

**Experience Assessment:**

I have read many papers in this area.

**Review Assessment: Checking Correctness Of Derivations And Theory:**

I assessed the sensibility of the derivations and theory.

**Review Assessment: Checking Correctness Of Experiments:**

I carefully checked the experiments.

**Review Assessment: Thoroughness In Paper Reading:**

I read the paper at least twice and used my best judgement in assessing the paper.

---

> ### Author Response · Authors · 2019-11-14
> **Response to Review #3**
>
> Thank you for taking the time to review our paper.
>
> 1- Re: large-scale experiment
> We ran a control experiment on the bowling Atari game using Impala (see Figure 7-c)  that tested whether the gains using R2D2 were not specific to Q-value based methods. These results suggest the benefits at scale are not limited to the value-based R2D2 setting. Testing the approach more broadly (on dmlab or challenging continuous control tasks as you suggested) is certainly something we want to look at in the future.
>
> 2- RE:sensitivity:
> Yes this is a good point. It is not overly sensitive to these exact values (a dimension of 16 for \phi does fine for example) but much larger values did tend to perform worse when we were tuning the architecture. One hypothesis is that a \phi with small dimensionality regularize the representation to only include relevant features, while larger dimensional \phi may contain less relevant information that will distract the modeling effort on phi. We plan to investigate that aspect more in future work.
>
> Thank you for the suggestions regarding the figures. We’ll include the learning curves for all the games in the appendix.
>
> We take your point about Figure 3, we’ll think about a way to make it more useful without relying too much on the text.

---

### Decision · Program_Chairs · 2019-12-19

**Decision:**

Reject

**Comment:**

This paper studies the problem of estimating the value function in an RL setting by learning a representation of the value function. While this topic is one of general interest to the ICLR community, the paper would benefit from a more careful revision and reorganization following the suggestions of the reviewers.